# Genotype-Phenotype Correlation in Familial *BAG3* Mutation Dilated Cardiomyopathy

**DOI:** 10.3390/genes13020363

**Published:** 2022-02-17

**Authors:** Karolina Mėlinytė-Ankudavičė, Marius Šukys, Jurgita Plisienė, Renaldas Jurkevičius, Eglė Ereminienė

**Affiliations:** 1Department of Cardiology, Medical Academy, Lithuanian University of Health Sciences, LT-50161 Kaunas, Lithuania; jurgita.plisiene@kaunoklinikos.lt (J.P.); renaldas.jurkevicius@lsmuni.lt (R.J.); eglerem@yahoo.com (E.E.); 2Department of Genetics and Molecular Medicine, Lithuanian University of Health Sciences, LT-50161 Kaunas, Lithuania; marius.sukys@kaunoklinikos.lt; 3Laboratory of Clinical Cardiology, Institute of Cardiology, Lithuanian University of Health Sciences, LT-50162 Kaunas, Lithuania

**Keywords:** *BAG3*, dilated cardiomyopathy, heart failure, inherited cardiomyopathies

## Abstract

We report the case of a 22-year-old male who visited a cardiologist after the first episode of atrial fibrillation (AF). Echocardiography and magnetic resonance imaging revealed decreased left ventricular (LV) systolic function with dilated LV. An intermittent second-degree AV (atrioventricular) block was detected during 24 h Holter monitoring. Genetic test revealed the pathogenic variant of the *BAG3* (BLC2-associated athanogene 3) gene. Due to the high risk of heart failure (HF) progression and ventricular arrhythmias, an event recorder was implanted and a pathogenetic HF treatment was prescribed. The analysis of genealogy revealed that the patient’s father, at the age of 32, was diagnosed with dilated cardiomyopathy (DCM) and recurrent AF episodes. Genetic testing also confirmed a pathogenic variant of the *BAG3* gene. Currently, with the optimal treatment of HF, the patient’s disease has been stable for three years and the condition is closely monitored on an outpatient basis. So, we demonstrate the importance of early detection for genetic testing and the unusual stability exhibited by the patient‘s optimal medical therapy for 3 years.

## 1. Introduction

Dilated cardiomyopathy (DCM) is defined by left ventricular (LV) systolic dysfunction and dilation in the absence of abnormal loading conditions that could explain myocardial abnormality [1]. This pathology can be subdivided into two main groups: genetic or acquired. Genetic testing should be performed in all individuals with a clinical diagnosis of cardiomyopathy. It was noticed that patients carrying malignant Lamin A/C gene (*LMNA*) variants were identified as first having a higher sudden cardiac death risk. Therefore, the implantation of an implantable cardioverter-defibrillator (ICD) has been found to be beneficial in these patients. [2] The HRS (Heart Rhythm Society) expert consensus document provided the recommendations for ICD placement in four genes related with an increased arrhythmic risk in moderately reduced LV systolic function. One of them is *BAG3* (BLC2-associated athanogene 3), which is a cochaperone that interacts with members of the heat shock protein family. Heterozygous *BAG3* pathogenic variants are rarely detected, they cause damage to the structure of myofibrils and are associated with impaired contractile function. The main risk factors for adverse outcomes in patients with DCM due to *BAG3* mutation are male sex, decreased LV systolic function, and LV dilatation [1,3]. The largest cohort of DCM caused by *BAG3* mutations revealed that this clinical situation is defined by early onset disease in most patients with a high risk of progression and a worse prognosis [1]. We discuss the case of a young man with an unusual genotype–phenotype correlation in familial dilated cardiomyopathy associated with the pathogenic *BAG3* variant.

## 2. Case Presentation

### 2.1. Clinical Presentation

A 22-year-old man was admitted to a cardiology outpatient clinic after a history of one episode of atrial fibrillation, which was restored by pharmacological and electrical cardioversion. He complained of palpitations, intermittent chest pain, and variation of arterial blood pressure (ABP). The patient was engaged in active sports and was taking supplements, such as proteins and amino acids. At physical examination, he had a heart rate of 70–80 bpm and ABP of 116/70 mmHg, no murmurs, and a normal sinus rhythm. Lung examination showed no alterations. His body mass index was 23.57 kg/m^2^ without peripheral edema (separate clinical assessments of pit depth and recovery at three locations were used (lower calf above the medial malleolus, behind the medial malleolus, and dorsum of the foot)).

The patient’s routine blood biochemistry, including electrolytes, renal function test, and complete blood count, was within normal limits, with normal B-type natriuretic peptide concentration (Table 1).

The ECG (electrocardiogram) was without significant rhythm and conduction disturbances (Figure 1a). However, an intermittent second-degree atrioventricular block during 24 h Holter monitoring was detected (Figure 1b).

Moreover, dilated LV with decreased systolic function and global longitudinal myocardial strain were found in cardiac 2D TTE (transthoracic echocardiography) (Figure 2). Magnetic resonance imaging did not find myocardial fibrosis by LGE (late gadolinium enhancement), T1 and T2 mapping, and confirmed changes found on echocardiography (Figure 3).

Two-dimensional TTE and Holter monitoring were repeated at follow up after 3 months of the discontinuation of active sports and protein intake. Persistently decreased LV systolic function was observed, with no evidence of arrhythmia or conduction abnormalities.

### 2.2. Genetic Test

The patient was referred for genetic testing, and the next-generation sequencing of 231 gene coding regions related with inherited heart disorders was performed. The test revealed heterozygous *BAG3* (NM_004281.3) variant c.514C > T causing premature STOP codon (Gln172Ter). This variant was interpreted as pathogenic, because nonsense variants are known to cause the disease, it is not found in population databases, and there is one submission in the ClinVar database classifying it as pathogenic.

Due to the high risk of progression of heart failure (HF) and ventricular arrhythmias, after the consultation with the geneticists, an electrophysiologist, the head of the Center for Rare Cardiovascular Diseases as well as international experts, an event recorder (Reveal™ XT 2529 Insertable Cardiac Monitor, Medtronic, Minneapolis, MN, USA) was implanted subcutaneusly, and a pathogenetic heart failure treatment was prescribed. The patient was treated with angiotensin-converting enzyme (perindopril 5 mg/per day) and β-blocker (bisoprolol 5 mg/per day). The close monitoring of the disease lasted for three years. The general condition was LV systolic function remained without changes, and no new rhythm or conduction disturbances are recorded during the event registrar’s examinations.

As the dilated cardiomyopathy caused by *BAG3* is inherited by an autosomal dominant pathway, genetic testing was offered to the first-degree relatives. The variant was found in the 51-year-old patient’s father, and he was referred to a cardiologist. Anamnesis revealed myocardial disease, which was later confirmed as DCM in another hospital, with repeated episodes of atrial fibrillation (AF), at 32 years of age. The closer investigation of genealogy (Figure 4) showed that the patient’s grandfather and his two sisters as well as brother died because of heart disease at the age of 60.

At present, the patient has been observed for 3 years after the established diagnosis; the size and function of the LV remains without changes, and no new rhythm or conduction disorders have been detected. The patient’s father has been followed for 19 years since the diagnosis of DCM, and optimal treatment for HF is currently being continued.

## 3. Discussion

DCM is a myocardial disease described as the dilatation and systolic impairment of the LV without abnormal loading conditions. The progression of a cardiomyopathic phenotype is related with a multiple interaction between cellular signaling pathways, stressors, and individual genotypes. Cardiac inflammation and fibrosis play an important role in the pathogenesis of DCM. Many cell types, such as neutrophils, monocytes, lymphocytes, macrophages, and others, are involved in the pathogenesis. Cardiac fibrosis occurs early in the progression of the disease, as a result of the persistent activation of cardiac inflammation, increasing cardiac rigidity and myocardial remodeling [4]. This disease varies from mild to severe, progressing to death [5], mostly diagnosed between 20 to 50 years old. The incidence of DCM is reported with 5–7 cases per 100,000 persons per year [6]. Early diagnosis, treatment, and follow-up of the disease have led to a significant improvement in the 5-year survival prognosis over the last 3 decades, ranging from 62% to 93% [7].

DCM is associated with more than 50 genes, but only 12 of them are causative of the disease, and have strong research-based evidence: *BAG3*, *DES*, *FLNC*, *LMNA*, *MYH7*, *PLN*, *RBM20*, *SCNSA*, *TNNC1*, *TNNT2*, *TTN*, and *DSP* [8]. The amount of genes associated with adverse outcomes among patients with DCM is growing. While the evidence base supporting a more malignant outcome is most convincing for DCM associated with variants in *LMNA*, it is also likely to be the case for disease caused by variants in *FLNC*, *DSP*, *BAG3*, *RBM20*, and *DES*. [9]. The detection of causative mutations immediately amplifies the possibilities for disease prevention through carrier screening and prenatal testing. Different gene mutations result in different course of the disease, and phenotypic variability is depending on the affected gene and the type of genetic variant as well as additional genetic or environmental factors. *TTN* (encoding titin protein) truncating mutations were reported to be a common cause of dilated cardiomyopathy. [10] *LMNA*-related DCM usually presents in adulthood, frequently accompanied by the significant conduction of system disease. According to the recommendations of the European Society of Cardiology (Class IIa level of evidence B), in the presence of pathogenic variants of the titin (TTN) gene, an ICD implantation is recommended for the primary prevention of sudden cardiac death [8]. One of the rarer genetic causes of DCM, as in the case we have described, is *BAG3* gene mutation, a newly discovered DCM-related gene, which is associated with about 2% of DCM patients [10,11]. *BAG3* is a heat shock protein (HSP) co-chaperone that induces degradation through autophagy [12]. It is determined that *BAG3* is essential for the normal production of filamin and is responsible of myocyte contraction [3]. This protein plays an important role in the heart, are released during cellular stress and their role is to stimulate the repair and degradation of protein aggregates. They are mainly produced in skeletal muscle cells and cardiomyocytes, and DCM-associated *BAG3* variations may promote apoptosis of myocytes [9]. Martin et al. presented results that indicate that *BAG3* is required for the functional maintenance of the cardiac sarcomere through mediating sarcomere protein turnover. Additionally, sarcomere dysfunction due to BAG3 haploinsufficiency arises despite normal sarcomere morphology [12]. The authors revealed that a DCM-associated *BAG3* mutation impaired *BAG3* binding to HSP70 and is related with the instability of HSPBs and protein aggregation [12]. New pathogenic variants of the *BAG3* gene correlate with very severe forms of the disease, especially at a young age, and in familial DCM [9]. This is confirmed in other studies; recently, a retrospective study showed that mutations in the *BAG3* gene due to DCM were mostly associated with early onset and rapid progression of the heart failure. Furthermore, the response to treatment in this gene mutation seemed less effective than on other genetic forms of DCM [3]. Therefore, our patient is particularly at an increased risk of adverse outcomes due to risk factors, such as male sex, young age, decreased LV systolic function, and LV dilatation. Moreover, familial DCM was confirmed. However, our clinical case shows the unusual disease stability with the combination of angiotensin converting enzyme inhibitors (ACEi) and β-blocker.

Unlike other DCM types, *BAG3*-associated DCM does not have significant arrhythmogenicity. In recent studies, worse outcomes were associated with the degree of LV dilatation and EF reduction as well as male sex. [13]. It was noticed that heat shock transcription factor-1 (HSF-1) expression in the myofilament fraction decreases in male patients with DCM and positively correlates with myofilament *BAG3* [14]. Although one of the main risk factors for poor prognosis in some studies is male gender [11,13], other studies show that dangerous arrhythmias did not occur more frequently in men than in women [3]. However, it is difficult to establish the disease chronology, as we see in our clinical case. Norton et al. described a wide range at the onset of the DCM (from 21 to 64 years old) and indicated high disease variability. There are no known data on the triggers for strengthening the gene expression, as in the situation we presented.

Thus, the results of most studies suggest that DCM caused by pathogenic variants of the *BAG3* gene detected at a young age is associated with rapid disease progression, life-threatening arrhythmias, poor prognosis, and survival. In our case, the patient was also young, 22 years of age at diagnosis, and an event recorder was implanted due to the risk of severe arrhythmias. Although our patient has been stable for three years, close monitoring is continued every year (dynamics of cardiac structure and function assessed by echocardiography and CMR at baseline and 1 year follow up are presented in Table 2) due to the high risk of sudden cardiac death.

Therefore, based on the described clinical case, early diagnosis of the disease, optimal treatment, and active monitoring are important, especially if we have familial DCM due to pathogenic *BAG3* gene variant; in this way, we can achieve better outcomes and treatment results.

## 4. Conclusions

DCM caused by pathogenic *BAG3* gene variant is defined by an aggressive clinical course dominated by HF complications and adverse prognosis. Poor outcomes are associated with male sex, low LV systolic function and increased LV diameter. With these risk factors, it is particularly important close monitor the patient and conduct optimal HF treatment.

## Figures and Tables

**Figure 1 genes-13-00363-f001:**
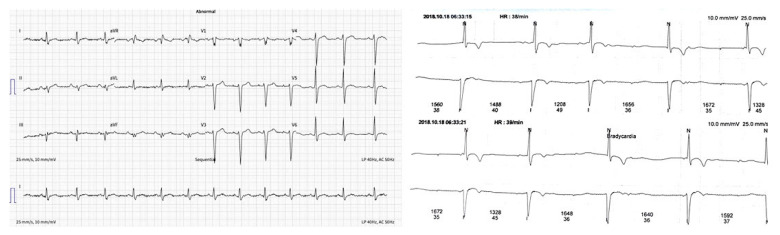
(**a**) ECG–without significant changes; (**b**) ECG–an intermittent second-degree AV block.

**Figure 2 genes-13-00363-f002:**
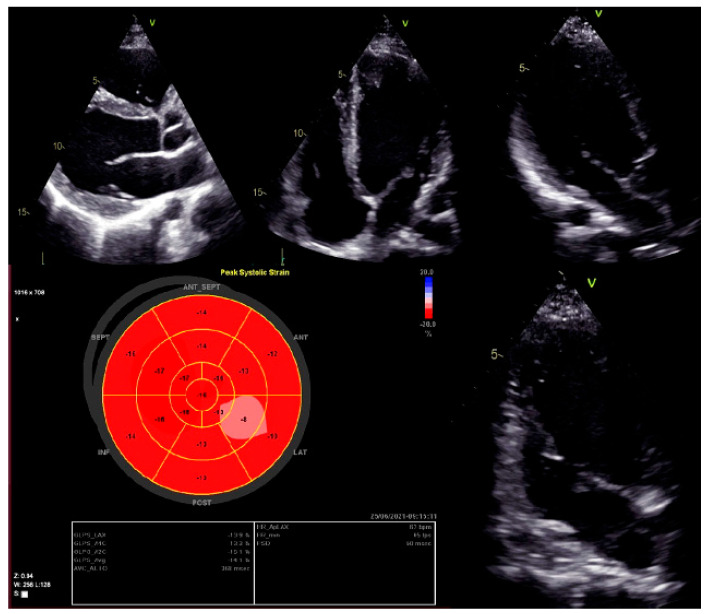
Left ventricular global longitudinal strain by speckle-tracking echocardiography.

**Figure 3 genes-13-00363-f003:**
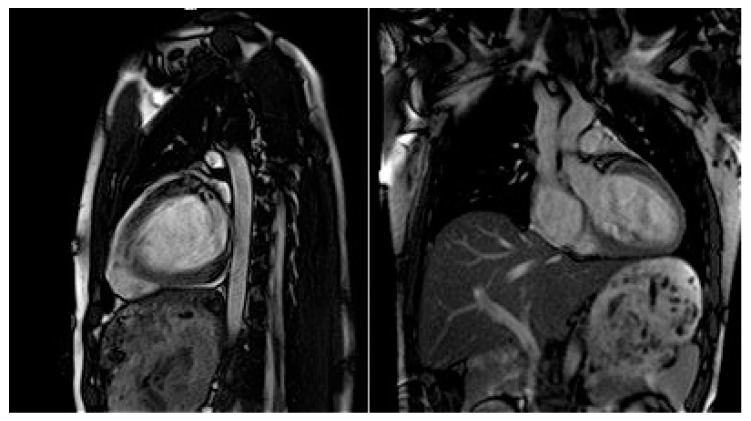
Cardiac magnetic resonance imaging with decreased LV systolic function and dilated LV.

**Figure 4 genes-13-00363-f004:**
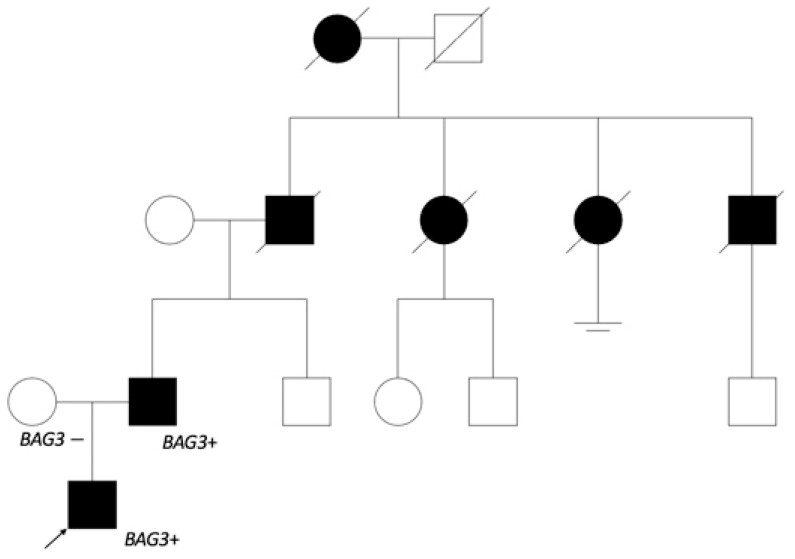
The patient’s genealogy scheme. The patient’s father started having heart insufficiency symptoms at 30 years of age. The grandfather from father’s side and his sisters died of heart disorder about 60 years of age, and the grandfather’s brother at 40 years of age. It is noted that grandfather’s sister started having arrhythmias at a younger age. The great-grandmother also died from a heart disorder. No relevant information from the patient’s mother’s side. Genetic testing was offered to the patient’s father’s brother and his cousins.

**Table 1 genes-13-00363-t001:** Clinical data.

Research	Description
Laboratory blood test	Within the normal range (creatinine 108 mcmol/L, K 5.5 mmol/L, Mg 0.85 mmol/L, RBC 5.16 × 10′12/L, WBC 9.74 × 10′9/L, HGB 160 g/L, PLT 274 × 10′9/L, BNP 4 ng/L).
ECG	Sinus rhythm, heart rate 80 bpm (Figure 1a).
Holter monitoring	Sinus rhythm, rare supraventricular extrasystoles, an intermittent second degree AV block, maximal R–R interval 1752 ms (Figure 1b).
2D TTE echocardiography	LVEDD 58 mm (LVEDDi 30.69 mm/m^2^), LVEDV 133 mL (LVEDVi 70.37 mL/m^2^), LV EF 45%, LV longitudinal strain −12%. RV 38 mm, RA 34 mm, RV longitudinal strain −19.4 proc. Conclusions: normal Ao valve morphology and function. LA non-dilated, normal MV function. Dilated LV, decreased contractility function of the posterolateral LV wall and global longitudinal LV myocardial strain. Decreased systolic LV function. RV size and function within normal limits.
Cardiac MRI	LVEDD 60 mm (LVEDDi 32.61 mm/m^2^), LVEDV 213 mL (LVEDVi 115.76 mL/m^2^), LV EF 43.2%. RV EF 50.8%. Conclusions: moderately dilated LV, decreased global systolic LV function. Preserved RV systolic function.

Ao—aortic; AV—atrioventricular block; BNP—B type natriuretic peptide; ECG—electrocardiogram; HGB—hemoglobin; EF—ejection fraction; LA—left atrium; LV—left ventricle; LVEDD—left ventricle end-diastolic diameter; LVEDDi—left ventricle end-diastolic diameter index; LVEDV—left ventricle end-diastolic volume; LVEDVi—left ventricle end-diastolic volume index; Mg—magnesium; MRI—magnetic resonance imaging; MV—mitral valve; RBC—red blood cell; RA—right atrium; RV—right ventricle; PLT—platelets; K—potassium; TTE—transthoracic echocardiography; WBC—white blood cell.

**Table 2 genes-13-00363-t002:** Dynamics of cardiac structure and function assessed by 2D echocardiography and CMR at baseline and after 1 year.

Parameters	Baseline	After 1 Year
BNP ng/L	4	4
2D echocardiography		
LVEDD (LVEDDi)	58 mm (30.69 mm/m^2^)	57 mm (30.89 mm/m^2^)
LVEDV (LVEDVi)	133 mL (70.37 mL/m^2^)	190 mL (103.26 mL/m^2^)
LVESV (LVESVi)	73 mL (38.62 mL/m^2^)	109 mL (59.23 mL/m^2^)
LV EF	45%	43%
LV longitudinal strain	−12%	−10%
RV	38 mm	38 mm
RA	34 mm	42 mm
RV longitudinal strain	−19.4%	−19%
Cardiac magnetic resonance		
LVEDD (LVEDDi)	61 mm (33.8 mm/m^2^)	60 mm(32.61 mm/m^2^)
LVEDV (LVEDVi)	213 mL (113 mL/m^2^)	213 mL (115.76 mL/m^2^)
LVESV (LVESVi)	124 mL (66 mL/m^2^)	121 mL (65.76 mL/m^2^)
LVSV (LVSVi)	89 mL (49.4 mL/m^2^)	92 mL (50 mL/m^2^)
LAA LAAi	20 cm^3^ (10.87 cm^3^)	20 cm^3^ (10.87 cm^3^)
LV EF	42%	43.2%
RV EF		50.8%

EF–ejection fraction; LV–left ventricle; LVEDD–left ventricle end-diastolic diameter; LVEDDi–left ventricle end-diastolic diameter index; LVEDV–left ventricle end-diastolic volume; LVEDVi–left ventricle end-diastolic volume index; LVESV–left ventricle end-systolic volume; LVESVi–left ventricle end-systolic volume index; LVSV–left ventricle systolic volume; LVSVi–left ventricle systolic volume index; LAA–left atrium area; MV–mitral valve; RA–right atrium; RV–right ventricle; LAAi–left atrium area índex; BNP–B-type natriuretic peptide.

## Data Availability

Not applicable.

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
