# Peer review of "Genotype-Phenotype Correlation in Familial *BAG3* Mutation Dilated Cardiomyopathy"

_genes, 2022, doi:10.3390/genes13020363_

Round 1
Reviewer 1 Report
The submitted case report divulges the early presentation of a male DCM patient with genetic link to BAG3 mutation. The correlation of such inherited causes of DCM are well established. This case should highlight the importance of early detection (22 years old vs. 40-50 avg) for genetic screening and the unusual stability exhibited by this medical managed (ACEi + beta blocker) patient for over 3 years. A slight adjustment to refocus on the differences for this case would increase the value added to the field.
Title: “Genotype–phenotype correlation in familial dilated cardiomyopathy due to BAG3 mutation – case report and review of the literature” The current title is overreaching, as this is not a true review (only 11 references). Suggest – “Genotype–phenotype correlation in familial BAG3 mutation dilated cardiomyopathy”
Examples of missing articles for a proper review:
- Domínguez F, Cuenca S, Bilińska Z, et al. Dilated Cardiomyopathy Due to BLC2-Associated Athanogene 3 (BAG3) Mutations. J Am Coll Cardiol. 2018;72(20):2471-2481. doi:10.1016/j.jacc.2018.08.2181
- Martin, T.G., Myers, V.D., Dubey, P. et al. Cardiomyocyte contractile impairment in heart failure results from reduced BAG3-mediated sarcomeric protein turnover. Nat Commun 12, 2942 (2021). https://doi.org/10.1038/s41467-021-23272-z
- Martin, T.G., Tawfik, S., Moravec, C.S., Pak, T.R., Kirk, J.A. BAG3 Expression and Sarcomere Localization in the Human Heart are Linked to HSF-1 and Are Differentially Affected by Sex and Disease. AM. J. Physiol. Heart Circ. 2021; 320(6): H2339-2350.
Line 16 – BAG3 not defined.
Line 18 – HF not defined.
Line 19 – Dilative = Dilated
Line 34 – Please define abbreviations upon first use. Abbreviations used in the abstract should be included in the introduction/body as needed. Ex: Heart Rhythm Society (HRS), LV, DCM, LGE
Line 47 – has = had
Line 52 –Standard International should be used units when reporting values: Ex: 23,57 = 23.57 (Change throughout).
Line 53 - How was peripheral edema measured?
Line 74 – Transthoracic (TTE) or transesophageal (TEE)?
Line 113 – DCMP? = DCM
Line 127 – “Detection of causative…”
Line 132 – Citation before the period. “DCM [9].”
Line 172 – Report states that patient was followed for 3 years post diagnosis, what about year 2 + 3 results for Table 2 or even the 3month followup from baseline? Statistics should be reported for comparison between baseline and year 1 results. What stats tests were used? Statistical significance >0.05?
Line 179 – Mg and RBC is not included in Table 2, remove.
Author Response
Dear reviewer,
thank you for your comments.
The title has been abbreviated as you recommend. The recommended benefits of this case are slightly highlighted.
Line 16 – BAG3 not defined
Corrected.
Line 18 – HF not defined
Corrected.
Line 19 – Dilative = Dilated
Corrected.
Line 34 – Please define abbreviations upon first use. Abbreviations used in the abstract should be included in the introduction/body as needed. Ex: Heart Rhythm Society (HRS), LV, DCM, LGE
Corrected. Thank you for your attention.
Line 47 – has = had
Corrected.
Line 52 –Standard International should be used units when reporting values: Ex: 23,57 = 23.57 (Change throughout)
Corrected.
Line 53 - How was peripheral edema measured?
It was explained in the text.
Line 74 – Transthoracic (TTE) or transesophageal (TEE)? ->
TTE. Corrected.
Line 113 – DCMP? = DCM
Corrected.
Line 127 – “Detection of causative…”
Corrected.
Line 172 – Report states that patient was followed for 3 years post diagnosis, what about year 2 + 3 results for Table 2 or even the 3month followup from baseline? Statistics should be reported for comparison between baseline and year 1 results. What stats tests were used? Statistical significance >0.05?
We didn't compare the statistics between baseline and follow up because these are single patient data and the numbers would be statistically insignificant.
Line 179 – Mg and RBC is not included in Table 2, remove
Corrected.
Sincerely,
Authors.
Reviewer 2 Report
The authors reported a case report of 22-year-old patient with DCM caused by BAG3 mutation. The case is well described and of clinical interest.
Comment:
- The authors did not provide a literature review. Thus, the title should be modified.
- In the discussion section, physiopathology of DCM could be better described (useful reference: https://doi.org/10.3390/ijms21186462)
Author Response
Dear reviewer,
thank you for your comments.
The title and discussion section have been modified as you recommend.
Sincerely,
Authors.